# Randomized Double-Blind Placebo-Controlled Study of Salivary Substitute with Enzymatic System for Xerostomia in Patients Irradiated in Head and Neck Region

**Letícia Pacheco Porangaba** [1], **Flávio de Melo Garcia** [1], **Ana Paula Alvarenga Antonio Rabelo** [2], **Amanda Puche Andrade** [2], **Fabio de Abreu Alves** [1,3] ID, **Antonio Cássio Assis Pellizzon** [4] ID **and Graziella Chagas Jaguar** [1,*]

[1]  Stomatology Department, A.C. Camargo Cancer Center, São Paulo 01509-010, Brazil;
    leticia.porangaba@accamargo.org.br (L.P.P.); flaviodemelogarcia@gmail.com (F.d.M.G.);
    falves@accamargo.org.br (F.d.A.A.)
[2]  Pradel Pharmacy, São Paulo 05592-140, Brazil; sorriso.nota10@yahoo.com.br (A.P.A.A.R.);
    amandapuche@hotmail.com (A.P.A.)
[3]  Stomatology Department, School of Dentistry, São Paulo University, São Paulo 05508-000, Brazil
[4]  Radiotherapy Department, A.C. Camargo Cancer Center, São Paulo 01509-010, Brazil;
    acapellizzon@accamargo.org.br
[*]  Correspondence: graziella.jaguar@accamargo.org.br

**Abstract:** This study aims to compare whether the use of a salivary substitute including an enzymatic system clinically reduces the intensity of xerostomia, as well as exploring the impact that this has on the quality of life, in patients who had received radiotherapy in the head and neck (HNC) region. Forty patients who had completed radiotherapy treatment within 6 months to 1 year previously were allocated into an Enzymatic Spray group (*n* = 21) or a Placebo arm (*n* = 19). It should be noted that two patients in the Placebo arm declined to participate during phase 2 of the study. All patients were randomized and used both products three times a day for 30 days. For analysis, xerostomia grade, unstimulated (UWS) and stimulated (SWS) salivary flow rate, and quality of life through the University of Washington Quality of Life Questionnaire validated in Portuguese (UW-QoL) were assessed in two phases: Phase 1 (before the use of the products) and Phase 2 (after 30 days of using the products). All clinical data were collected from medical records. Analyzing the salivary substitute with the enzymatic system, an improvement in xerostomia complaints was observed 30 days after using the product; however, this difference was not statistically significant (*p* > 0.05). Regarding quality of life, no significant differences were observed in relation to the UW-QoL and saliva domain between the groups in the two phases of the study (*p* > 0.05). The salivary substitute with the enzymatic system may be effective in reducing radio-induced xerostomia symptoms; however, further research is necessary to evaluate the efficacy of this salivary substitute on oral health.

**Keywords:** xerostomia; hyposalivation; salivary substitute; head and neck cancer; radiotherapy; parotid gland; radiation injuries

## 1. Introduction

Radiotherapy (RT) is widely used for the treatment of head and neck cancer (HNC), either exclusively or associated with chemotherapy (CT) and/or surgery. Despite the effectiveness and technological advances in planning and techniques used for RT, side effects are common in the oral cavity [1,2].

Xerostomia (dry mouth) and hyposalivation are oral complications in patients undergoing RT for HNC. The inclusion of salivary glands in the radiation fields (with doses between 2 and 10 Gy) can lead to decreases in their function by about 50–60% within the first week of treatment [3,4]. In addition, RT interferes with the composition of saliva, which, as a result, becomes more viscous. As such, the patient may present with changes in

physiological functions, as well as an increased risk of opportunistic infection and dental caries [5].

Radiation caries progresses more rapidly and is associated with a greater risk of dental treatment failure, related to severe de-mineralization [5]. For this reason, it is recommended that patients who have or will undergo RT maintain regular oral care, allowing for the early identification of carious lesions and fluoride applications [6]. Most recent technologies using biomimetic materials are the result of research efforts focused on the use of tissue engineering technology to shift from enamel re-mineralization to enamel re-generation. Biomimetic hydroxyapatite-based toothpastes have been investigated due to their re-mineralizing activity on dental surfaces, reducing hypersensitivity/pain values more effectively than conventional fluoride toothpaste [7].

The literature has presented a range of different methods for the management of radio-induced xerostomia and relieving associated symptoms, including the use of cholinergic agents, photobiomodulation, salivary substitutes, acupuncture, intra-oral electrostimulation, and stem cell therapies [8–10]. It is well known that the solution to xerostomia may not reside in a single approach but, rather, in the use of a combination of agents [11]. A multidisciplinary approach to the management of xerostomia is considered essential. Among all available therapies, salivary substitutes are well known. The Bioxtra Spray® is a salivary substitute, which differs from other products, presenting an enzymatic system in its composition and allowing for balancing the microbiota in the oral cavity [12,13].

In the past, salivary substitutes were based on carboxymethylcellulose and hydroxyethylcellulose. In addition to offering limited lubrication, they do not promote protection of the oral cavity. Thus, a new generation of salivary substitutes has emerged over the years, including complex formulations simulating the salivary peroxidase system, as well as salivary components, such as lysozymes, lactoferrins, and some immunoglobulins [14–16].

Thus, considering the possible advantages associated with the use of Bioxtra Spray®, the aim of this study was to assess whether the use of this product is capable of clinically reducing xerostomia intensity, thus positively impacting the quality of life of patients who are undergoing IMRT or RTC3D in HNC.

## 2. Materials and Methods

### 2.1. Characterization of the Study

A randomized, double-blind, and placebo-controlled study considering the use of Bioxtra Spray® was conducted in patients with radiation-induced xerostomia. The study was approved by the local Ethics Committee of the A.C. Camargo Cancer Center, Sao Paulo, Brazil (no.2543/18). All patients provided written informed consent for their inclusion in this study.

### 2.2. Inclusion Criteria

- Patients between the ages of 18 and 75 with oropharynx or nasopharynx squamous cell carcinoma (clinical stage ≥ II) that had been treated with RTC3D or IMRT (≥45 Gy) in the bilateral cervico–facial region.
- Patients who subjectively complained of xerostomia 6–12 months after the completion of RT.

### 2.3. Exclusion Criteria

Patients with previous history of RT and/or surgery in the head and neck region, patients who were taking cholinergic drugs, and patients who did not attend all phases of the study were excluded.

### 2.4. Patient Recruitment

Eligible ambulatory patients were recruited consecutively at the Stomatology Department at the A.C. Camargo Cancer Center in Sao Paulo, Brazil, from September 2019 to September 2020.

### 2.5. Study Design

This study encompassed two arms: (1) Bioxtra Spray® and (2) Placebo. All patients were allocated using institutional sequential allocation software and were stratified considering the following variables: concomitant chemotherapy and radiotherapy technique (RTC3D or IMRT). All parotid glands were contoured using anatomical atlas, and corresponding doses in the irradiated volumes of 25%, 50%, 75%, and 100% of each gland were acquired using a dose–volume histogram (DVH).

The physician was informed of the individual patient codes, and two coded spray bottles were given to the patients. All patients were enrolled between March 2019 and August 2020.

The estimated sample size was 40 cases divided into 2 groups, considering a 93% confidence interval and 5% significance level.

### 2.6. Treatment Protocol

Patients were instructed to spray the products (i.e., Bioxtra Spray® and Placebo) twice in the oral cavity three times a day for 30 days. Both products met the specifications recommended by the Brazilian Health Regulatory Agency (ANVISA) regarding their organoleptic and physical–chemical characteristics [17,18]. The formulations of both products are detailed below (Table 1).

**Table 1.** Bioxtra Spray® and Placebo formulations.

| BIOXTRA SPRAY® | | PLACEBO |
|---|---|---|
| -Water<br>-Sorbitol<br>-Maltitol<br>-Xylitol<br>-Hydroxyethylcellulose<br>-Sodium Benzoate<br>-Sodium Methylparaben<br>-Citric acid<br>-Sodium Chloride<br>-Propylbaraben Sodium | -Dipotassium phosphate<br>-Sodium Saccharin<br>-Calcium Chloride<br>-Magnesium Chloride<br>-Bovine Colostrum<br>-Lactoperoxidase<br>-Fluorine | -Water<br>-Xylitol<br>-Hydroxyethylcellulose |

### 2.7. Data Collection

Clinicopathological data were collected from the medical records of patients. All patients were evaluated by the same trained and blinded physician in two phases:

Phase 1: Before administration of spray;

Phase 2: 30 days after stopping the spray.

To analyze the rationality of self-medication, both the Bioxtra Spray® and Placebo bottles were weighed prior to being given to patients and again upon return.

### 2.8. Xerostomia Assessment

Xerostomia grade was assessed using observer-based grade and score according to the subjective measures of Eisbruch et al. [19]: grade 1, defined as dry mouth without interference in habits; grade 2, dry mouth with frequent ingestion of fluid to swallow; grade 3, dry mouth with impact on diet, sleep, speech, or other activities.

### 2.9. Salivary Flow Rate Evaluation

Whole unstimulated saliva (UWS) and stimulated saliva (SWS) flows were collected in the morning (between 8 a.m. and 11 a.m.) to minimize the circadian effects. Immediately before the test, the patient swallowed any saliva in the mouth and then started to expectorate the saliva that was spontaneously produced into a plastic funnel connected to a plastic tube. The SWS was collected using a parafilm® as a mechanical stimulus. This collection process took 10 min: the patients were instructed to chew the parafilm for 10 min and then

started to expectorate the saliva that was produced into another plastic funnel that was also connected to a plastic tube. The samples were then weighed, and the salivary flow rate was calculated and adjusted to ml/min [20].

### 2.10. Quality of Life Assessment

A Brazilian-Portuguese version of the University of Washington Quality of Life Questionnaire (UW-QOL) was given to all patients [21]. The questionnaire includes 12 items evaluating the domains of pain, appearance, activity, recreation, chewing, swallowing, speech, shoulder, taste, saliva, humor, and anxiety, where a value of 100 indicates the best level of overall function. There is a general agreement that a composite score of 75–100 has little effect on QOL.

### 2.11. Statistical analysis

Statistical analysis was performed using a software program (IBM SPSS for Windows version 25; SPSS Inc., Chicago, IL, USA). A descriptive analysis of the results was performed. Wilcoxon, *t*-test, Fisher, Mann–Whitney, and Kruskal–Wallis tests were performed in order to evaluate the statistical significance of observed differences between the groups. In all hypothesis tests, the level of significance was set at 5%.

## 3. Results

A total of 40 consecutive patients met the inclusion criteria, where 21 were assigned to the Bioxtra Spray® group and 19 to the Placebo group (Table 2). Of these 40 randomized patients, 2 in the Placebo arm declined to participate during phase 2 of the study. With this, 38 patients were retained for analysis. The trial flow diagram is shown in Figure 1. Based on the weight of the bottles, the adherence to drug treatment involving continuous use of the medications was noted.

**Table 2.** Clinical characteristics of the 40 head and neck cancer patients.

| Variables | Category | Bioxtra Spray® (*n* = 21) | Placebo (*n* = 19) | *p* |
|---|---|---|---|---|
| Age (years) | Mean ± SD | 60.09 ± 8.99 | 59.94 ± 11.80 | 0.469 |
| | Range | 40–72 | 30–69 | |
| | Median | 62 | 60 | |
| Gender | Male | 18 (85.7%) | 16 (84.2%) | 0.594 |
| | Female | 3 (14.3%) | 3 (15.8%) | |
| Tumor Site | Orapharynx | 19 (90.4%) | 16 (84.2%) | 0.195 |
| | Nasopharynx | 3 (9.5%) | 3 (15.7%) | |
| Clinical Stage | II | 9 (42.9%) | 6 (31.6%) | 0.376 |
| | III | 11 (52.4%) | 9 (47.4%) | |
| | IV | 1 (4.8%) | 4 (21.1%) | |
| Treatment | RT | 2 (9.5%) | 2 (10.5%) | 0.658 |
| | RT + CT | 19 (90.5%) | 17 (87.5%) | |
| Type of RT | RTC3D | 9 (42.9%) | 5 (26.3%) | 0.333 |
| | IMRT | 12 (57.1%) | 14 (73.7%) | |
| Total Dose (Gy) | ≤60 Gy | 0 (0.0%) | 0 (0.0%) | 0.282 |
| | De 60 a 70 Gy | 1 (4.7%) | 1 (5.2%) | |
| | >70 Gy | 20 (95.2%) | 18 (94.7%) | |
| Mean dose of RT in billateral parotid glands | Mean ± SD | 44.20 ± 19.0 | 39.74 ± 17.40 | 0.390 |
| | Median | 47.60 | 29.0 | |

D = standard deviation; RT = radiotherapy; SUR = surgery; CT = chemotherapy; RTC3D = three-dimensional conformal radiotherapy; IMRT = intensity-modulated radiation therapy.

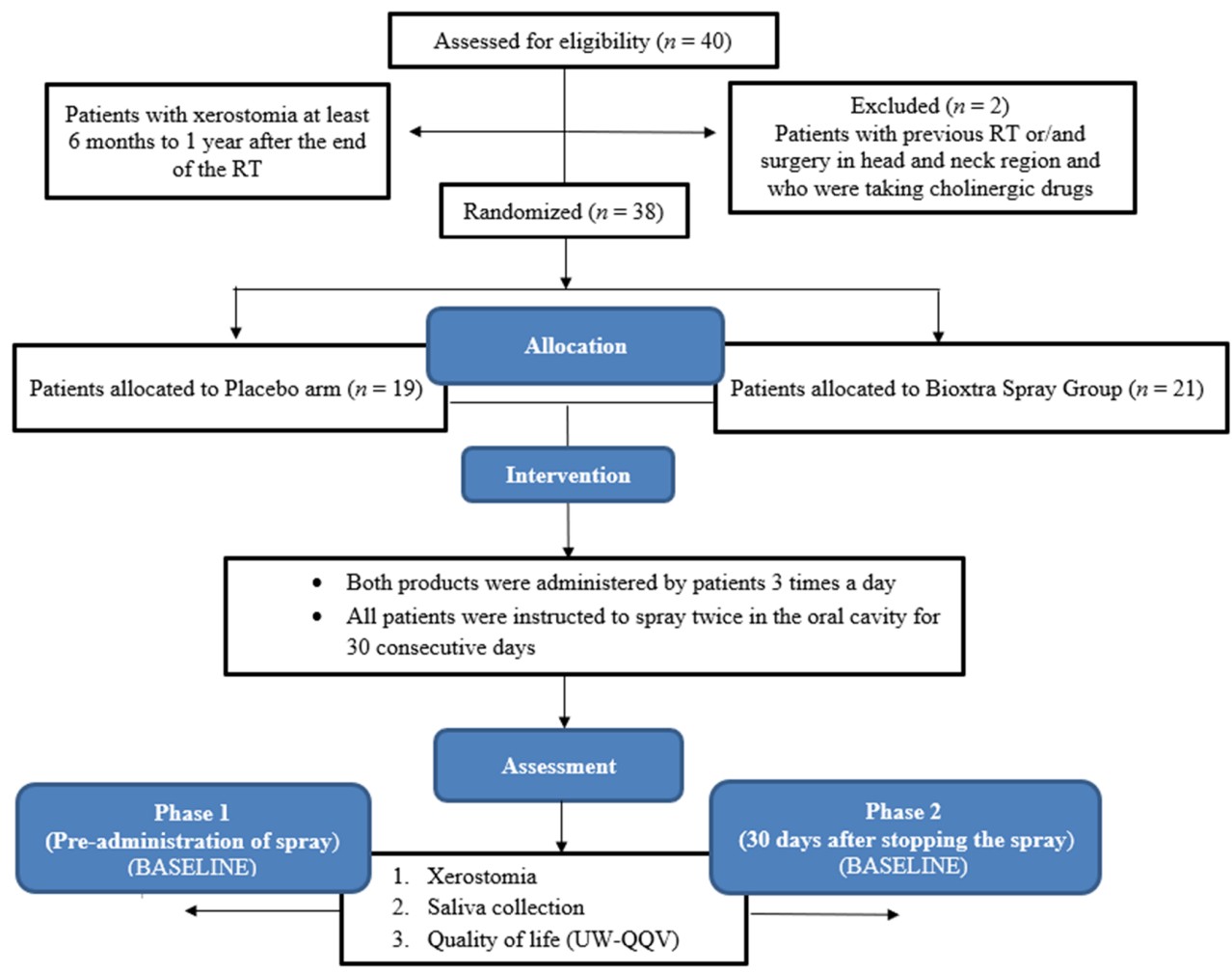

**Figure 1.** CONSORT diagram.

Baseline demographics and disease characteristics are presented in Table 2. Patients and treatment features were similar among the treatment groups.

### 3.1. Parotid Glands DVH

With regard to DVH, the mean dose in 75% of the parotid gland volume was significantly higher in patients who underwent RTC3D, in comparison to patients who underwent IMRT. We did not observe statistical differences between the groups ($p > 0.05$; Table 3). In this study, the sub-mandibular glands could not be spared due to their close proximity to level II neck nodes, which are usually included in the target volume.

**Table 3.** Effects of mean dose (Gy) and RT technique on parotid gland volume for both groups.

| Salivary Gland | Volume | Placebo Arm | | | Bioxtra Spray® Group | | |
| --- | --- | --- | --- | --- | --- | --- | --- |
| | | Mean Dose (Gy) RTC3D | Mean Dose (Gy) IMRT | $p^1$ | Mean Dose (Gy) RTC3D | Mean Dose (Gy) IMRT | $p^2$ |
| Rigtht Parotid | 25% | 70.50 | 43.27 | 0.003 * | 52.73 | 44.60 | 0.018 * |
| | 50% | 65.14 | 28.62 | 0.003 * | 49.15 | 38.72 | 0.095 |
| | 75% | 55.41 | 21.22 | 0.002 * | 40.25 | 21.61 | 0.041 * |
| | 100% | 77.45 | 69.94 | 0.007 * | 69.95 | 67.97 | 0.754 |

**Table 3.** *Cont.*

| Salivary Gland | Volume | Placebo Arm | | | Bioxtra Spray® Group | | |
|---|---|---|---|---|---|---|---|
| | | **Mean Dose (Gy) RTC3D** | **Mean Dose (Gy) IMRT** | $p^1$ | **Mean Dose (Gy) RTC3D** | **Mean Dose (Gy) IMRT** | $p^2$ |
| Left Parotid | 25% | 69.46 | 44.72 | 0.010 * | 66.04 | 46.98 | 0.003 * |
| | 50% | 64.85 | 27.37 | 0.001 * | 57.06 | 31.24 | 0.002 * |
| | 75% | 55.58 | 19.26 | 0.001 * | 47.19 | 26.40 | 0.018 * |
| | 100% | 73.04 | 65.41 | 0.087 | 71.45 | 70.08 | 0.651 |

$p^1$ = comparison between the mean dose (Gy) and RT RTC3D and IMRT in parotid gland volume in Placebo arm; $p^2$ = comparison between the mean dose (Gy) and RT RTC3D and IMRT in parotid gland volume in Bioxtra Spray® Group. * This symbol is to highlight the statistical significance.

### 3.2. Xerostomia Complaint

In phase 1, most patients in the Bioxtra Spray group had already reported some degree of xerostomia, being Grade 2 (*n* = 11 patients, 52.4%) or Grade 3 (*n* = 10 patients, 47.6%). Comparing patients in the same group 30 days after treatment, we observed a decrease in patients with Grade 3 xerostomia (*n* = 7 patients, 33.3%) and an increase in patients with grade 1 xerostomia (*n* = 2, 9.5%; *p* > 0.05; Table 4).

**Table 4.** Assessment of xerostomia grade.

| Phases | Grade of Xerostomia | Bioxtra Spray® Group | | $p^1$ | Placebo Arm | | $p^2$ | $p^3$ |
|---|---|---|---|---|---|---|---|---|
| | | *n* | % | | *n* | % | | |
| PHASE 1 | Grade 1 | 0 | (0.0%) | | 1 | (5.2%) | | |
| | Grade 2 | 11 | (52.4%) | | 9 | (47.4%) | | 0.870 |
| | Grade 3 | 10 | (47.6%) | | 9 | (47.4%) | | |
| | Total | 21 | (100%) | 0.753 | 19 | (100%) | 0.988 | |
| PHASE 2 | Grade 1 | 2 | (9.5% | | 1 | (5.9%) | | |
| | Grade 2 | 12 | (57.1%) | | 11 | (64.7%) | | 0.998 |
| | Grade 3 | 7 | (33.3%) | | 5 | (29.4%) | | |
| | Total | 21 | (100%) | | 17 | (100%) | | |

$p^1$ = comparison between the Bioxtra spray group grade xerostomia; $p^2$ = comparison between the Placebo arm grade xerostomia; $p^3$ = comparison between both groups and grade xerostomia.

Assessing the complaint of xerostomia in the Placebo arm, we observed that, in phase 1, most patients had Grade 2 (*n* = 9 patients, 47.4%) or Grade 3 (*n* = 9 patients, 47.4%) xerostomia. After 30 days of treatment, most patients remained at Grade 2 (*n* = 11 patients, 64.7%). No statistical differences were observed (*p* > 0.05; Table 4).

We did not observe any statistical differences between the groups (*p* > 0.05; Table 4).

### 3.3. Salivary Flow

In phase 1, we observed a mean SWS of 0.00 mL/min in patients in the Bioxtra Spray® group undergoing RTC3D, compared with a mean of 0.03 mL/min in patients who underwent IMRT (*p* = 0.023). Interestingly, these significant differences were also noted in phase 2 (SWS; RTC3D: 0.01 mL/min versus 0.04 mL/min IMRT; *p* = 0.009). No significant differences were observed in the Placebo arm (Table 5).

**Table 5.** Salivary flow rate (mL/min) UWS and SWS by RT technique between the groups.

| Phases | Salivary Flow Rate | Bioxtra Spray® Group | | $p^1$ | Placebo Arm | | $p^2$ | $p^3$ |
| | | RTC3D Mean (SD) | IMRT Mean (SD) | | RTC3D Mean (SD) | IMRT Mean (SD) | | |
|---|---|---|---|---|---|---|---|---|
| PHASE 1 | UWS | 0.00 (0.00) | 0.00 (0.01) | 0.069 | 0.00 (0.02) | 0.00 (0.00) | 0.754 | 0.098 |
| PHASE 2 | UWS | 0.00 (0.00) | 0.02 (0.05) | 0.545 | 0.00 (0.00) | 0.00 (0.01) | 0.591 | 0.545 |
| PHASE 1 | SWS | 0.00 (0.00) | 0.03 (0.05) | 0.023 * | 0.00 (0.01) | 0.01 (0.02) | 0.754 | 0.086 |
| PHASE 2 | SWS | 0.01 (0.00) | 0.04 (0.07) | 0.009 * | 0.00 (0.01) | 0.02 (0.030 | 0.591 | 0.076 |

$p^1$ = comparison between the Bioxtra spray group UWS and SWS in RTC3D and IMRT; $p^2$ = comparison between the Placebo arm UWS and SWS in RTC3D and IMRT; $p^3$ = comparison between both groups and UWS and SWS and RTC3D and IMRT; SD = standard deviation; RTC3D = three-dimensional conformal radiotherapy; IMRT = intensity-modulated radiation therapy; UWS = whole unstimulated saliva; SWS = whole stimulated saliva. * This symbol is to highlight the statistical significance.

### 3.4. Quality of Life Assessment

At baseline, the mean global quality of life score did not present any statistically significant difference between the groups. Comparing the 12 UW-QoL domains, the saliva domain presented the worst score: 37.2 for the Placebo arm and 26.8 for the Bioxtra Spray group ($p > 0.05$).

After treatment, the global mean scores for both groups increased; however, the Bioxtra Spray group obtained a lower score of 76.0, when compared to the Placebo arm with 85.7 ($p < 0.05$). We did not observe any significant difference between the groups regarding the saliva domain (41.6 for the placebo group and 30.0 for the Bioxtra Spray® group; $p > 0.05$; Table 6).

**Table 6.** Mean of 12 UW-QOL domains in both groups for the two phases.

| Domain | PHASE 1 | | | PHASE 2 | | |
| | Placebo | Bioxtra Spray Group | $p^1$ | Placebo | Bioxtra Spray Group | $p^2$ |
|---|---|---|---|---|---|---|
| 1. Pain | 94.7 | 87.5 | 0.569 | 98.5 | 90.4 | 0.486 |
| 2. Apprearance | 87.5 | 80.9 | 0.349 | 91.1 | 80.9 | 0.068 |
| 3. Activity | 91.6 | 77.3 | 0.119 | 96.8 | 88.0 | 0.254 |
| 4. Recreation | 88.8 | 78.5 | 0.294 | 96.8 | 80.9 | 0.032 * |
| 5. Chewing | 72.6 | 66.9 | 0.504 | 73.0 | 68.5 | 0.465 |
| 6. Swallowing | 72.2 | 73.8 | 0.878 | 80.0 | 78.5 | 0.899 |
| 7. Speech | 94.5 | 90.5 | 0.530 | 97.9 | 93.7 | 0.514 |
| 8. Shoulder | 90.7 | 93.4 | 0.965 | 100.0 | 90.0 | 0.211 |
| 9. Taste | 70.9 | 47.5 | 0.030 * | 73.0 | 44.4 | 0.010 * |
| 10. Saliva | 37.2 | 26.8 | 0.486 | 41.6 | 30.0 | 0.370 |
| 11. Humor | 86.1 | 81.2 | 0.460 | 90.6 | 84.5 | 0.596 |
| 12. Anxiety | 88.2 | 76.3 | 0.045 * | 93.8 | 81.7 | 0.336 |
| Mean Score | 82.7 | 73.1 | 0.014 * | 85.7 | 76.0 | 0.018 * |

$p^1$ = comparison between the Placebo arm and Bioxtra spray group in phase 1; $p^2$ = comparison between the Placebo arm and Bioxtra spray group in phase 2. * This symbol is to highlight the statistical significance.

### 4. Discussion

The severity of radiation-induced xerostomia and hyposalivation is directly correlated with the mean dose of RT in the salivary glands. Several studies have shown that sparing the parotid and submandibular glands using techniques such as IMRT can minimize radiation damage to the salivary glands [22–25]. Nevertheless, none of our patients had

their submandibular gland spared during treatment planning, which may explain the reason for the intense xerostomia observed in both groups. The submandibular glands contribute 65–90% of unstimulated saliva, which is rich in salivary mucins responsible for oral lubrification. Over the past few decades, authors have reached the conclusion that xerostomia can be substantially reduced through limiting the maximum mean dose threshold to 39 Gy for at least one submandibular gland [26]. A potential disadvantage is the possible locoregional recurrence. For this reason, it must be indicated in selected patients, such that tumor control is not compromised [27,28].

In order to promote comfort and quality in patients with xerostomia after RT, the literature has provided several management regimes [8–11]. Among the forms of management, salivary substitutes and their various available formulations have been widely discussed [20–23]. These salivary substitutes have similar physical constituents as human saliva [29].

According to the literature, a few studies have used the Bioxtra system, which differs from the other available salivary substitute systems as it contains peptides and immunoglobulins that complete the mouth's natural antibacterial and immunological mechanism [12,13,30,31]. In this context, our study is the first randomized, double-blind, and Placebo-controlled study to evaluate the effectiveness of Bioxtra Spray®.

During the period of our study, patients used both products over a period of 30 consecutive days. Shahdad et al. [12] evaluated the effectiveness of the Bioténe salivary substitute system compared to the Bioxtra® system, where patients used the products for two consecutive weeks. Dirix et al. [30], in order to evaluate the Bioxtra® salivary substitute system, had patients use the product for four weeks; meanwhile, Bakhshi et al. [13] evaluated the effectiveness of Bioxtra Spray® and mouthwash for four weeks. Therefore, the ideal time for using such products has not yet been defined in the existing literature.

In our study, although we did not use formulations of salivary substitute systems which differ according to the mode of application, in comparison with the Placebo arm, we did not observe any significant differences in relation to the xerostomia grade after 30 days of applying the Bioxtra Spray® three times a day ($p = 0.796$).

Bakhshi et al. [13] compared the effectiveness of Bioxtra Spray® and mouthwash in terms of relieving xerostomia in irradiated patients at least 6 months after the end of treatment. However, no significant differences were observed in symptom relief between groups. Similarly, we did not observe significant differences in symptom relief between groups. However, analyzing the patients who used Bioxtra Spray®, we observed an interesting improvement in the intensity of xerostomia 30 days after using the product.

Based on the literature, no studies on the effectiveness of the Bioxtra® salivary substitute system (or similar) have evaluated the salivary flow rate through sialometry of the UWS and SWS [12,13,30,31]. Salivary substitutes are not expected to improve the salivary flow rate after RT damage, due to the nature of their properties being only in relation to the lubrication and protection of the soft and hard tissues of the oral cavity. However, the patients in this study who used Bioxtra Spray® curiously presented a significantly greater difference in both UWS and SWS, in relation to the Placebo arm, 30 days after using the product.

This occurrence can be clarified through the literature, where authors believe that some recovery of the function of the salivary glands can occur over time, especially in cases where the IMRT technique is performed. It has been shown that the remaining intact salivary gland stem cells and/or progenitor cells could determine the regenerative capacity of the salivary gland after IMRT [32,33].

Using these more precise RT techniques, it was possible to verify the radiation dose distribution at 25%, 50%, 75%, and 100% of the volume of the parotid glands in both groups and correlate it with the RT technique. Corroborating the literature, we observed that the patients in this study who underwent IMRT in both groups received significantly lower mean doses in all volumes of the parotid glands when compared to patients who underwent 3DRTC. In the present study, despite all our efforts to optimize the irradiation dose to the

parotid glands, the majority of patients who underwent IMRT reported grade 2 and/or 3 xerostomia in phase 1. Notably, there was no statistical difference after using the Bioxtra spray®. Furthermore, we also found no differences when comparing the xerostomia results in patients who underwent IMRT with those who underwent 3DRTC. These data once again emphasize the importance of the submandibular glands in the role of xerostomia.

Regarding the overall difference in quality of life between the groups, the results indicated that, in both phases of the study, the mean for the Placebo arm was 82.5 in phase 1 and 85.7 in phase 2, while the means in the Bioxtra Spray® group were 73.1 and 76.0, respectively, with statistical differences observed between the two phases of the study ($p < 0.05$). Dirix et al. [30] observed an improvement in the average quality of life score, being 59.4 before treatment with Bioxtra moisturizing gel, while, after 28 days of using the product, the mean quality of life score increased to 70.5.

Analyzing the QQV-UW domains separately, in phase 1 of the study, we observed scores of 37.2 for the Placebo arm and 26.8 for the Bioxtra Spray® group in the saliva domain. In phase 2 (still in relation to the saliva domain), an improvement was observed in both scores, with 41.6 for the Placebo arm and 30.0 for the Bioxtra Spray® group. However, no statistical difference was observed ($p > 0.05$). For comparison, Shahdad et al. [12] observed that the Bioxtra® salivary substitute system achieved significantly better rates for xerostomia and improvements in speech and chewing when compared with the Biotène® line ($p < 0.05$). One important reason may be the fact that the follow-up period was too short to detect any discernable differences in our patients.

Based on our findings, although we did not observe significant differences between the two groups, Bioxtra Spray® seems to play an important role in the reduction of xerostomia and improvement in salivary flow, having an impact on the quality of life of the patient. Further studies with larger sample sizes and applying the product for a longer period are required to validate the results obtained in this study. To clarify the role of Bioxtra® spray in the protection of oral tissues, more trials focused on its action in the oral microbiota are necessary. In addition, we also highlight the importance of focusing efforts on the prevention of xerostomia through the consideration of strategies, such as submandibular gland sparing through protection against radiation (e.g., IMRT), intraoral stents, and the use of preventive sialogogues.

**Author Contributions:** Conceptualization: F.d.M.G., G.C.J., A.C.A.P., F.d.A.A. and L.P.P.; methodology: G.C.J. and L.P.P.; validation: G.C.J., A.C.A.P. and L.P.P.; formal analysis: G.C.J., A.C.A.P. and L.P.P.; investigation: G.C.J. and L.P.P.; resources: A.P.A.A.R. and A.P.A.; data curation: L.P.P.; writing—original draft preparation: L.P.P. and G.C.J.; writing—review and editing: L.P.P. and G.C.J.; visualization: all authors; supervision: G.C.J., A.C.A.P. and F.d.A.A. All authors have read and agreed to the published version of the manuscript.

**Funding:** This research received no external funding.

**Institutional Review Board Statement:** The study was conducted according to the guidelines of the Declaration of Helsinki and approved by the Institutional Ethics Committee of the A.C. Camargo Cancer Center, Sao Paulo, Brazil (no. 2543/18), registration number of trial (REBEC): (RBR: 9sdf3k).

**Informed Consent Statement:** Informed consent was obtained from all subjects in the study. Written informed consent was obtained from the patients to publish this paper.

**Data Availability Statement:** Data supporting the reported results can be found at https://dados.accamargo.org.br/redcap/redcap_v13.1.27/index.php?pid=271.

**Acknowledgments:** The authors would like to thank CAPES (Coordenação de Aperfeiçoamento de Pessoal de Nível Superior) for the financial support. We would also like to thank Pradel Pharmacy (São Paulo, Brazil) for all their support throughout the research and Bioxtra (FNL comercio de Suprimentos Ltd.a- Rio de Janeiro, Brazil).

**Conflicts of Interest:** None of the authors have a financial relationship with the organization that sponsored the research. All authors have full control of all primary data and agree to allow the journal to review their data if requested.

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
