# Peer review of "Randomized Double-Blind Placebo-Controlled Study of Salivary Substitute with Enzymatic System for Xerostomia in Patients Irradiated in Head and Neck Region"

_curroncol, doi:10.3390/curroncol31020082_

Round 1

Reviewer 1 Report

Comments and Suggestions for Authors

This study shows that Bioxtra Spray® is effective against xerostomia, and this topic is very interesting. Because, xerostomia is one of the leading side effects after radiotherapy of head and neck cancer and is difficult to treat. However, there are some questions and points to be supplemented as follows;

Clarify the interval to initial application of oral spray in each group (Bioxtra spray® group vs. placebo group) after the completion of radiation therapy.

According to the QUANTEC criteria, xerostomia can occur in both salivary glands based on mean dose of 20-25 Gy. In addition, it is also affected by irradiated volume of each glands. Therefore, it is important to clarify that there is no difference between the two groups by describing the mean dose and the volume that receives 20-25 Gy of parotid glands and submandibular glands, respectively.

Additionally, description of the radiation field is needed. (eg. both cervical lymphatic levels)

Comments on the Quality of English Language

 Minor editing of English language required

Author Response

Reviewer, thank you very much for your considerations. We read calmly so that we can adjust in the best way possible. Bellow are your comments answered.

  • Clarify the interval to initial application of oral spray in each group (Bioxtra spray® group vs. placebo group) after the completion of radiation therapy.

Answer: We clarified that the interval to initial application of oral spray in both group were at least 6 months after the completion of radiation therapy. We added in Abstract section: “ Forty patients who had completed at least 6 months of HNC were allocated in: Enzimatic Spray group (n =21) and Placebo arm (n=19).” This was also considered as our inclusion criteria.

  • According to the QUANTEC criteria, xerostomia can occur in both salivary glands based on mean dose of 20-25 Gy. In addition, it is also affected by irradiated volume of each glands. Therefore, it is important to clarify that there is no difference between the two groups by describing the mean dose and the volume that receives 20-25 Gy of parotid glands and submandibular glands, respectively.

Answer: We added in Study Design section the sentence: “All parotid glands were contoured using anatomical atlas and corresponding dose in the irradiated volumes of 25%, 50%, 75% and 100% of each gland was acquired using a dose–volume histogram (DVH).” We also clarified the mean dose in parotid glands in both groups according to RT techinique (Table 3)

Table 3Mean dose (Gy) and technique of RT between the groups in parotids glands volume.

Salivary

Gland

Volume

Placebo Arm

Bioxtra Spray® Group

Mean Dose

(Gy)

RTC3D

Mean Dose (Gy)

IMRT

p¹

Mean Dose

(Gy)

RTC3D

Mean Dose

(Gy)

IMRT

p2

Rigtht Parotid

25%

70.50

43.27

0.003 *

52.73

44.60

0.018 *

50%

65.14

28.62

0.003 *

49.15

38.72

0.095

75%

55.41

21.22

0.002 *

40.25

21.61

0.041 *

100%

77.45

69.94

0.007 *

69.95

67.97

0.754

Left Parotid

25%

69.46

44.72

0.010 *

66.04

46.98

0.003 *

50%

64.85

27.37

0.001 *

57.06

31.24

0.002 *

75%

55.58

19.26

0.001 *

47.19

26.40

0.018 *

100%

73.04

65.41

0.087

71.45

70.08

0.651

† p1= Comparison between the mean dose (Gy) and RT RTC3D and IMRT in parotid gland volume in Placebo arm;  p2= Comparison between the mean dose (Gy) and RT RTC3D and IMRT in parotid gland volume in Bioxtra Spray® Group.

Parotid Glands DVH

With regard to DVH, the mean dose in 75% of the parotid glands volume was significantly higher in patients who underwent RTC3D in comparison to patients who underwent IMRT. We did not observe statistical differences between the groups (p>0.05) (Table 3). In this study, the submandibular glands could not be spared because of their close proximity to level II neck nodes, usually included in the targets volume.

  • Additionally, description of the radiation field is needed. (eg. both cervical lymphatic levels)

Answer: We added the description of radiation fied in the Inclusion criteria: “Patients were between the age of 18 and 75 with oropharynx or nasopharynx squamous cell carcinoma, clinical stage > II that had been treated with RTC3D or IMRT (>45Gy) with the bilateral cervico-facial region”

Reviewer 2 Report

Comments and Suggestions for Authors

Thank you for the opportunity to review this paper about xerostomia management. 

This paper would benefit from English proof reading to increase the readability. 

Abstract

-       Grammar ‘to compared’ should be ‘to compare’

-       Grammar ‘impacting on the quality of life’ should be ‘and explore the impact that this has’

-       Please note the grading system for xerostomia

-       Spelling ‘dispite’ should be ‘despite’

-       Should note that allocation was randomised

Introduction

-        Phrasing for second paragraph. “When the salivary glands are included in the radiation fields, their function decreases by about 50-60% already in the first week of treatment, with doses between 2-10 Gy” needs revision

-       Would be good to elaborate on the difference in composition between bioxtra spray and other saliva substitutes (elaborating on paragraph 3) given that is the intervention of interest

Methods

-       Needs some more work on the syntax and grammar throughout

-       Inclusion criteria – this reads as the results, ie the range that was recruited, but was it also the inclusion criteria? Ie did you exclude anyone over the age of 75? If so why?

-       Study design – ‘total of 40 patients’ this belongs in the results. 

-       Good outline of the composition of the two treatments (table works well). Is the placebo current standard of care or is it expected to not have any effect?

-       Figure 1 belongs in the results

Results 

-       How many patients were screened for suitability? Was this 40 and hence none excluded? Or were there some not eligible, and if so why not? This is important to understand if there is selection bias in your sample.

-       Did you look at adherence? I wonder if a variable that is not accounted for were patients who did not do the 3 sprays vs some who perhaps found it helpful and did more than 3 sprays per day. 

-       Table 2: either here or in the results you need to put time in relation to their treatment. ie did they all start the spray at the beginning of their treatment or was it more adhoc than this. I wonder if those who are during RT vs those who are some time post respond to the spray differently

-       Baseline – was this pre-radiation? Or was it pre-administration of spray? And regardless what was the median start time for all patients (and range)

Discussion 

-       First sentence – yes directly related to radiation dose but you should also note that it is worse for multimodal treatment where theyhave had a parotidectomy first. 

-       Please elaborate on why the addition of peptides and immunoglobulins are important for this population. 

-       I think you should rethink the 5th paragraph. Some of the previous literature would do better in the introduction to provide background and evidence to the bioxtra but that there was a limitation in the types of study (ie no RCT) hence your aim is to assess in RCT. Then in the discussion you could reference the same papers but only in relation to there being a range of soray duration (3 – xxx weeks)

-       Please provide exact p values

-       When you say you didn’t notice any difference between groups, was your study powered adequately to detect this?

-       Did the PROM ratings go up when the UWS and SWS improved? Would be interesting to see whether the scores are associated with each other

Conclusion 

-       You note that there is a combination of therapies used to improve xerostomia, but this is the first time this is mentioned. Xerostomia management by the MDT should be incorporated into the introduction. 

Comments on the Quality of English Language

requires review re syntax and grammar throughout

Author Response

Reviewer, thank you very much for your considerations. We read calmly so that we can adjust in the best way possible. Bellow are your comments answered.

Abstract

The gramar of the abstract was adjusted

-       Grammar ‘to compared’ à ‘to compare’

-       Grammar ‘impacting on the quality of life’ à ‘and explore the impact that this has’

-       Spelling ‘dispite’ à be ‘despite’

-       Should note that allocation was randomised

Introduction

-       Would be good to elaborate on the difference in composition between bioxtra spray and other saliva substitutes (elaborating on paragraph 3) given that is the intervention of interest

Answer: We added the sentence :

               In the past, salivary substitutes were based on carboxymethylcellulose and hydroxyethylcellulose. In addition to offering limited lubrication, they did not promote protection of the oral cavity. Thus, a new generation of salivary substitutes has emerged over the years with complex formulations simulating the salivary peroxidase system, but also introduce salivary components such as lysozymes, lactoferrins and some immunoglobulins [14-16].

 -        Phrasing for second paragraph. “When the salivary glands are included in the radiation fields, their function decreases by about 50-60% already in the first week of treatment, with doses between 2-10 Gy” needs revision

Answer: We added the sentence : alivary glands included in the radiation fields, can decrease its function by about 50-60% already in the first week of treatment, with doses between 2-10 Gy [3,4].

Methods

-       Needs some more work on the syntax and grammar throughout

-       Inclusion criteria – this reads as the results, ie the range that was recruited, but was it also the inclusion criteria? Ie did you exclude anyone over the age of 75? If so why?

Answer: The age was a inclusion criteria :

Patients were between the age of 18 and 75 with oropharynx or nasopharynx squamous cell carcinoma

-       Study design – ‘total of 40 patients’ this belongs in the results. –

ok

-       Good outline of the composition of the two treatments (table works well). Is the placebo current standard of care or is it expected to not have any effect?

            Answer: The placebo was expected to not have any effect. However, the results show that a single lubrification of the oral mucosa is positive with respect to xerostomia. However, the durability of the placebo seems to be less than Bioxtra Group.

Results 

-       How many patients were screened for suitability? Was this 40 and hence none excluded? Or were there some not eligible, and if so why not? This is important to understand if there is selection bias in your sample.

Answer: A total of 40 patients were randomized, of whom 21 were assigned to Bioxtra Spray® group and 19 to Placebo (Table 2). From these 40 randomized patients, 2 in the Placebo arm declined to participate during phase 2 of the study. Therefore, in phase 2, the results of 38 patients were analyzed.

-       Did you look at adherence? I wonder if a variable that is not accounted for were patients who did not do the 3 sprays vs some who perhaps found it helpful and did more than 3 sprays per day. 

Answer: To analyze the rationality of self-medication, both Bioxtra Spray® and Placebo bottles were weighed prior to being given to patients and weighed again upon return.

-       Table 2: either here or in the results you need to put time in relation to their treatment. ie did they all start the spray at the beginning of their treatment or was it more adhoc than this. I wonder if those who are during RT vs those who are some time post respond to the spray differently

Answer: Patients who subjectively complained of xerostomia with at least 6 months to 1 year after the end of the RT.

-       Baseline – was this pre-radiation? Or was it pre-administration of spray? And regardless what was the median start time for all patients (and range)

Phase 1: Pre-administration of spray

Phase 2: with 30 days after stopping the spray

Discussion 

-       First sentence – yes directly related to radiation dose but you should also note that it is worse for multimodal treatment where theyhave had a parotidectomy first. 

The severity of radiation-induced xerostomia and hyposalivation are directly correlated with the mean dose of RT in the salivary glands. 

-       Please elaborate on why the addition of peptides and immunoglobulins are important for this population. 

Answer: These salivary substitutes have similar physical constituents of human saliva

-       I think you should rethink the 5th paragraph. Some of the previous literature would do better in the introduction to provide background and evidence to the bioxtra but that there was a limitation in the types of study (ie no RCT) hence your aim is to assess in RCT. Then in the discussion you could reference the same papers but only in relation to there being a range of soray duration (3 – xxx weeks)

ok

-       Please provide exact p values

ok

-       When you say you didn’t notice any difference between groups, was your study powered adequately to detect this?

               One important reason may be the fact that the follow up for our patients was too short to detect any discernable differences.

-       Did the PROM ratings go up when the UWS and SWS improved? Would be interesting to see whether the scores are associated with each other

            The PROM rating did not correlated to salivar flow.

Conclusion 

-       You note that there is a combination of therapies used to improve xerostomia, but this is the first time this is mentioned. Xerostomia management by the MDT should be incorporated into the introduction. 

It is well known that the solution to xerostomia may not reside in a single approach but rather in the use of a combination of agents (Jaguar et al 2017). A multidisciplinary team approach to managing xerostomia is considered essencial.

Reviewer 3 Report

Comments and Suggestions for Authors

A manuscript of considerable interest to the dental sector, of help both to colleagues in the field and to all professionals involved in the care of frail patients.

Needs major revision before evaluation for possible publication.

Abstract: remove the commercial name of the product while leaving the active ingredient, and emphasising the results obtained

Keywords: few, add specific ones recorded on MeSH.

Very poor introduction, natural substances such as ozone, ozonised water, laser and postbiotic-based gels, paraprobiotic-based mousse to reduce inflammation and increase salivary flow should be included (Scribante et al)

Materials and methods: well described, but the sample size calculation is missing, and the consorto flow chart not well drafted, please use the official format and make it readable.

Results: very confusing, reorganise the tables, and highlight the stastically significant results so that they stand out to the reader.

Discussion; add as a future goal the analysis of all natural substances that can also prevent hard tissue injuries, such as a biomimetic zinc-substituted hydroxypatite toothpaste (Butera et al.)https://doi.org/10.3390/ijerph19148676

Conclusions; add proactive pre-treatment action

Bibliography; add references required.

Author Response

Reviewer, thank you very much for your considerations. We read calmly so that we can adjust in the best way possible. Bellow are your comments answered.

A manuscript of considerable interest to the dental sector, of help both to colleagues in the field and to all professionals involved in the care of frail patients.

Needs major revision before evaluation for possible publication.

Abstract: remove the commercial name of the product while leaving the active ingredient, and emphasising the results obtained

Enzimatic Spray group

Keywords: few, add specific ones recorded on MeSH.

Keywords:  Xerostomia; Hyposalivation; Salivary substitute; Head and neck cancer; Radiotherapy; Parotid Gland; Radiation Injuries

Very poor introduction, natural substances such as ozone, ozonised water, laser and postbiotic-based gels, paraprobiotic-based mousse to reduce inflammation and increase salivary flow should be included (Scribante et al)

The literature presents different methods to manage radio-induced xerostomia, relieving symptoms such as cholinergic agents, photobiomodulation, salivary substitutes, acupuncture, intra-oral electrostimulation and  stem cell therapies         

Materials and methods: well described, but the sample size calculation is missing, and the consorto flow chart not well drafted, please use the official format and make it readable.

ok 

Results: very confusing, reorganise the tables, and highlight the stastically significant results so that they stand out to the reader.

 ok

Discussion; add as a future goal the analysis of all natural substances that can also prevent hard tissue injuries, such as a biomimetic zinc-substituted hydroxypatite toothpaste (Butera et al.)https://doi.org/10.3390/ijerph19148676

Radiation caries progress more rapidly and are also associated with a greater risk of dental treatment failure, related to severe demineralization (AUTOR). With this, it is recommended that patients who have or will undergo RT maintain a regular oral care, which will allow early identification of carious lesions and fluoride applications. Most recent technologies, using Biomimetic materials are the result of research efforts to use the tissue engineering technology to shift from enamel remineralization to enamel regeneration.  Biomimetic hydroxyapatite-based toothpastes have been investigated for their remineralizing activity on dental surfaces with reduction of hypersensitivity/pain values higher than conventional fluoride toothpaste (Butera et al.).

Round 2

Reviewer 2 Report

Comments and Suggestions for Authors

Abstract

-       Line 20; does not make sense. Currently reads as they had 6 months to 1 year of cancer, however I suspect you mean they are 6 months to 1 year post cancer treatment

-       I am still uncertain as to the recruitment strategy, I understand that 40 were recruited but was this consecutively? How many did not meet inclusion criteria or withdrew? 

-       Prospective/retrospective

-       Sentence line 25-27 needs grammatical revision. 

Introduction 

-       Line 45 should read ‘patient may present with…’

-       Line 50 should read ‘maintain regular oral care’

-       Line 87 should read ‘complained of xerostomia 6 – 12 months after the completion of RT’

Method

-       Lin 116 should read ‘within 30 days…’

Results 

-       As per methods I am still unclear on the numbers these 40 were selected from. Was it consecutively from an outpatient clinic referral base?

-       Line 165 – should read RTC3D compared with 

Comments on the Quality of English Language

another review regarding the grammar would be of benefit

Author Response

Firstly, we would like to apologize for the delay in sending the revised paper. We work exhaustively for its improvement. We also expanded the words and revised the text. Unfortunately, upon rereading the English revised manuscript, we found that we attached the Manuscript with an error in the fifth paragraph of the discussion and missing 2 references, as follows: Correct Paragraph: “This occurrence can be clarified through the literature, where authors believe that some recovery of the function of the salivary glands can occur over time, especially in cases where the IMRT technique is performed. It has been shown that the remaining intact salivary gland stem cells and/or progenitor cells could determine the regenerative capacity of the salivary gland after IMRT [32-33].”

Missing References: 32. van Luijk P, Pringle S, Deasy JO, Moiseenko VV, Faber H, Hovan A, Baanstra M, van der Laan HP, Kierkels RG, van der Schaaf A, Witjes MJ, Schippers JM, Brandenburg S, Langendijk JA, Wu J, Coppes RP. Sparing the region of the salivary gland containing stem cells preserves saliva production after radiotherapy for head and neck cancer. Sci Transl Med. 2015 Sep 16;7(305):305ra147.

  1. Hawkins PG, Lee JY, Mao Y, Li P, Green M, Worden FP, Swiecicki PL, Mierzwa ML, Spector ME, Schipper MJ, Eisbruch A. Sparing all salivary glands with IMRT for head and neck cancer: Longitudinal study of patient-reported xerostomia and head-and-neck quality of life. Radiother Oncol. 2018 Jan;126(1):68-74. doi: 10.1016/j.radonc.2017.08.002. Epub 2017 Aug 16.

Once again we ask for forgiveness and we would like to know how we can resolve this issue since we are no longer able to change the manuscript?

-       I am still uncertain as to the recruitment strategy, I understand that 40 were recruited but was this consecutively? How many did not meet inclusion criteria or withdrew? 

 Answer: We better clarified the patient recruitment in the method section, and also, added more details on discontinued and retained patients, as follow:

“Patient recruitment Eligible ambulatory patients were recruited consecutively at the Stomatology Departament at A.C. Camargo Cancer Center in São Paulo, Brazil, from February 2020 to October 2022.”

Reviewer 3 Report

Comments and Suggestions for Authors

The manuscript has been properly revised and can be published

Author Response

Firstly, we would like to apologize for the delay in sending the revised paper. We work exhaustively for its improvement. We also expanded the words and revised the text. Unfortunately, upon rereading the English revised manuscript, we found that we attached the Manuscript with an error in the fifth paragraph of the discussion and missing 2 references, as follows: Correct Paragraph: “This occurrence can be clarified through the literature, where authors believe that some recovery of the function of the salivary glands can occur over time, especially in cases where the IMRT technique is performed. It has been shown that the remaining intact salivary gland stem cells and/or progenitor cells could determine the regenerative capacity of the salivary gland after IMRT [32-33].”

Missing References: 32. van Luijk P, Pringle S, Deasy JO, Moiseenko VV, Faber H, Hovan A, Baanstra M, van der Laan HP, Kierkels RG, van der Schaaf A, Witjes MJ, Schippers JM, Brandenburg S, Langendijk JA, Wu J, Coppes RP. Sparing the region of the salivary gland containing stem cells preserves saliva production after radiotherapy for head and neck cancer. Sci Transl Med. 2015 Sep 16;7(305):305ra147.

  1. Hawkins PG, Lee JY, Mao Y, Li P, Green M, Worden FP, Swiecicki PL, Mierzwa ML, Spector ME, Schipper MJ, Eisbruch A. Sparing all salivary glands with IMRT for head and neck cancer: Longitudinal study of patient-reported xerostomia and head-and-neck quality of life. Radiother Oncol. 2018 Jan;126(1):68-74. doi: 10.1016/j.radonc.2017.08.002. Epub 2017 Aug 16.

Once again we ask for forgiveness and we would like to know how we can resolve this issue since we are no longer able to change the manuscript?